# Extracting relevant predictive variables for COVID-19 severity prognosis: An exhaustive comparison of feature selection techniques

**Miren Hayet-Otero** [1,2,3º], **Fernando García-García** [1º*], **Dae-Jin Lee** [1,4],
**Joaquín Martínez-Minaya** [5], **Pedro Pablo España Yandiola** [6], **Isabel Urrutia Landa** [7],
**Mónica Nieves Ermecheo** [7,8], **José María Quintana** [8], **Rosario Menéndez** [9], **Antoni Torres** [10],
**Rafael Zalacain Jorge** [11], **Inmaculada Arostegui** [1,12], **with the COVID-19 & Air Pollution
Working Group** [¶]

**1** Basque Center for Applied Mathematics (BCAM), Bilbao, Basque Country, Spain, **2** Department of
Electronic Technology, University of the Basque Country (UPV/EHU), Leioa, Basque Country, Spain,
**3** Basque Research and Technology Alliance (BRTA), TECNALIA, Derio, Basque Country, Spain, **4** School of
Science and Technology, IE University, Madrid, Madrid, Spain, **5** Department of Applied Statistics and
Operational Research, and Quality, Universitat Politècnica de València (UPV), Valencia, Valencian
Community, Spain, **6** Respiratory Service, Galdakao-Usansolo University Hospital, Galdakao, Basque
Country, Spain, **7** BioCruces Bizkaia Health Research Institute, Barakaldo, Basque Country, Spain,
**8** Research Unit, Galdakao-Usansolo University Hospital, Galdakao, Basque Country, Spain, **9** Pneumology
Department, La Fe University and Polytechnic Hospital, Valencia, Valencian Community, Spain,
**10** Pneumology Department, Hospital Clínic of Barcelona, Barcelona, Catalonia, Spain, **11** Pneumology
Service, Cruces University Hospital, Barakaldo, Basque Country, Spain, **12** Department of Mathematics,
University of the Basque Country (UPV/EHU), Leioa, Basque Country, Spain

☯ These authors contributed equally to this work.
¶ Membership of the COVID-19 & Air Pollution Working Group is listed in the Acknowledgments.
* fegarcia@bcamath.org

doi.org/10.1371/journal.pone.0284150

University, KOREA, REPUBLIC OF

**Data Availability Statement:** The dataset which
supports the findings of this study could not be
made publicly available due to restrictions from our

## Abstract

With the COVID-19 pandemic having caused unprecedented numbers of infections and
deaths, large research efforts have been undertaken to increase our understanding of the
disease and the factors which determine diverse clinical evolutions. Here we focused on a
fully data-driven exploration regarding which factors (clinical or otherwise) were most infor-
mative for SARS-CoV-2 pneumonia severity prediction via machine learning (ML). In partic-
ular, feature selection techniques (FS), designed to reduce the dimensionality of data,
allowed us to characterize which of our variables were the most useful for ML prognosis. We
conducted a multi-centre clinical study, enrolling $n = 1548$ patients hospitalized due to
SARS-CoV-2 pneumonia: where 792, 238, and 598 patients experienced low, medium and
high-severity evolutions, respectively. Up to 106 patient-specific clinical variables were col-
lected at admission, although 14 of them had to be discarded for containing ⩾60% missing
values. Alongside 7 socioeconomic attributes and 32 exposures to air pollution (chronic and
acute), these became $d = 148$ features after variable encoding. We addressed this ordinal
classification problem both as a ML classification and regression task. Two imputation tech-
niques for missing data were explored, along with a total of 166 unique FS algorithm configu-
rations: 46 filters, 100 wrappers and 20 embeddeds. Of these, 21 setups achieved
satisfactory bootstrap stability (⩾0.70) with reasonable computation times: 16 filters, 2 wrap-
pers, and 3 embeddeds. The subsets of features selected by each technique showed

Ethics Committees for Clinical Research (CEIc), as it contains potentially identifying or sensitive patient information. For further requests, please contact CEIc at hgu.ceic@osakidetza.eus.

**Funding:** This research is supported by the Spanish State Research Agency AEI under the project S3M1P4R PID2020-115882RB-I00, as well as by the Basque Government EJ-GV under the grant 'Artificial Intelligence in BCAM' 2019/00432, under the strategy 'Mathematical Modelling Applied to Health', and under the BERC 2018–2021 and 2022–2025 programmes, and also by the Spanish Ministry of Science and Innovation: BCAM Severo Ochoa accreditation CEX2021-001142-S / MICIN / AEI / 10.13039/501100011033. The funders had no role in study design, data collection and analysis, decision to publish, or preparation of the manuscript.

**Competing interests:** The authors have declared that no competing interests exist.

modest Jaccard similarities across them. However, they consistently pointed out the importance of certain explanatory variables. Namely: patient's C-reactive protein (CRP), pneumonia severity index (PSI), respiratory rate (RR) and oxygen levels –saturation Sp O2, quotients Sp O2/RR and arterial Sat O2/Fi O2–, the neutrophil-to-lymphocyte ratio (NLR) – to certain extent, also neutrophil and lymphocyte counts separately–, lactate dehydrogenase (LDH), and procalcitonin (PCT) levels in blood. A remarkable agreement has been found *a posteriori* between our strategy and independent clinical research works investigating risk factors for COVID-19 severity. Hence, these findings stress the suitability of this type of fully data-driven approaches for knowledge extraction, as a complementary to clinical perspectives.

# Introduction

## Motivation

The outbreak of the COVID-19 pandemic, caused by the SARS-CoV-2 virus, has brought unprecedented numbers of infections, severe affectations and deaths worldwide. Since the early stages of the pandemic, extraordinary research efforts have been carried out to understand the course of the disease, and to define effective evidence-based prevention and therapeutic guidelines. Such task has proven to be notably difficult, since the clinical evolution of COVID-19 is known to vary to a considerable extent across patients: from very mild affectation, to critical deterioration or death.

## Related works

Among those research efforts, a plethora of statistical and machine learning (ML) algorithms have been proposed in the literature to support medical decision-making for COVID-19 (see [1–3] for exhaustive reviews): in diagnosis, prognosis, assessment of the risk of hospitalization/death, or to counsel therapeutic management for an effective response against the disease.

With respect to the statistical approaches, Cecconi *et al.* [4] were among the first in developing a prognostic tool for clinical deterioration, using a multivariable Cox model. With the goal of predicting disease progression, a broad number of authors have employed diverse ML algorithms, such as: *k*-nearest neighbors, logistic regression, support vector machines (SVM), multi-layer perceptron neural networks, decision trees and random forest, or boosting techniques, among many others (e.g. [5–7]). Specifically, Varzaneh *et al.* [8] evaluated various of the classical ML algorithms in terms of their ability to predict patients' need for intubation due to an adverse progression of COVID-19. In a similar manner, several models have been proposed to predict mortality risk: Caillon *et al.* [9] used a penalized logistic regression to estimate the probability of death, as well as a regularized Cox regression model to predict survival. In addition, different authors have also addressed the performance of various ML strategies for mortality prognosis (e.g. [10, 11]).

Of note, the aforementioned studies reported about the importance of the features in their data (which is not so widespread when focusing on prediction capabilities). Different strategies were followed in order to identify the most relevant features: various works employed statistical hypothesis testing (e.g. [4]), correlation with the target variable and importance estimates available within the trained ML algorithm (e.g. [6, 11]). Embedded methods (such as $L^1$-penalised models) were used by [5, 9]. Wrapper methods, such as backward elimination with leave-

one-out and stepwise feature selection integrated with leave-one-out or k-fold validation, were used by Kocadagli *et al.* [7]. Interestingly, these authors also presented a novel wrapper methodology based on genetic algorithms and information complexity. Besides, Karthikeyan *et al.* [10] proposed a wrapper-based procedure, with a neural network as internal model for assessing feature importance, alongside an external XGBoost classification model.

However, since these works were primarily devoted to the prediction on COVID-19 clinical outcomes, they lack exhaustive analyses on the technique for assessing feature relevance. Instead, feature selection (FS) became just another step in their ML prediction pipelines, to circumvent the classical 'curse of dimensionality' issue when coping with high-dimensional datasets. For example, Varzaneh *et al.* carried out a comparison of six meta-heuristics for FS [8], although their resulting subsets of features were evaluated in terms of fitness, classification accuracy and number of selected features. Hence, reports on FS properties beyond predictive ability were omitted.

## Objective

In this context, we deemed suitable to complement the analyses in terms of prediction capabilities by ML owing to FS. Considering that there exists a broad range of FS algorithms of different nature, here we opted for conducting an explicit and systematic comparison about the behaviour of the various FS algorithms with respect to their robustness. Indeed, this topic constitutes a field of growing importance within the ML community for understanding the FS procedure itself [12, 13], where different methods have been proposed to enhance the assessments of the stability properties of FS concerning changes in the data (see [14] for a comprehensive methodological review). In this regard and to the best of our knowledge, the approach presented here stands aside from the existing literature, as we have not identified other studies tackling the topic of robustness in FS for COVID-19 data.

Furthermore, our approach was conceived as a data-driven exploration for factors which may serve as the most informative predictors for ML-enabled SARS-CoV-2 pneumonia severity prognosis. Remarkably, ML is consolidated as a prominent methodology for knowledge extraction from data in medical research [15], and complementary to clinical perspectives.

From a practical perspective, FS may also become beneficial: it may reduce remarkably the demands for data acquisition by healthcare professionals, since fewer variables imply ameliorating the labour-intensive and resource-consuming task of collecting clinical information. Furthermore and contrarily to other dimensionality reduction techniques –such as feature extraction techniques (e.g. projection via principal component analysis: PCA)–, FS has the inherent advantage of maintaining the original representation of the data unaltered, thus fostering the interpretability by human domain experts (here pulmonologists) on the chosen subset of features [16].

In addition, in our COVID-19 & Air Pollution Working Group we are also interested in studying the socioeconomic and environmental determinants of health. For this reason, we complemented patients' demographic and clinical variables with information about: *a*) their socioeconomic status (by postcode of residence, as a proxy for personal socioeconomic status), and *b*) their exposure to air pollutants (also by postcode). Not in vain, there exists increasing research and evidence about the effect of air pollution on the susceptibility to COVID-19 infection, severity and death [17–23]; as well as about demographic [24] and socioeconomic risk factors [25–29].

## Materials & methods

### Clinical data collection

Our COVID-19 & Air Pollution Working Group conducted an observational, retrospective, longitudinal, cohort study with a multi-centre setup, in four hospitals from three different geographical territories in Spain—One in Catalonia: Clínic Hospital (servicing urban/metropolitan Barcelona), one in the Valencian Community: La Fe Hospital (metropolitan Valencia), and two in the Basque Country: Galdakao-Usansolo and Cruces Hospital (respectively semi-urban/rural and metropolitan areas). The study was approved by the corresponding Ethics Committees for Clinical Research (reference codes: HCB/2020/0273, 20–122-1, PI 2019090, PI 2020083), and carried out in adherence to the relevant guidelines and regulations. Only participants who voluntarily gave written informed consent were enrolled.

The inclusion criterion was adult patients ($\geq$18 years old) admitted to in-hospital stays due to SARS-CoV-2 pneumonia during the first epidemic wave of COVID-19 in Spain: between mid-February and the end of May 2020. Requirements for SARS-CoV-2 pneumonia diagnosis were: a positive microbiological test (positive DNA amplification test by PCR for SARS-CoV-2), as well as compatible chest imaging findings (radiography and/or tomography).

*A posteriori* examinations of patients' electronic records allowed us to allocate cases by their actual clinical severity experienced. Our pulmonologists at the Respiratory Service of the Galdakao-Usansolo University Hospital defined three severity levels (low, medium and high) and systematic criteria for each. Further details can be found elsewhere [30].

A wide set of clinical variables were collected to describe each case. These included *a*) demographics (age, sex, body mass index, etc.); *b*) pre-existing comorbidities; *c*) physiological status; *d*) examinations at the time of hospitalization (blood analytics, arterial gas tests, etc.) [30]. To guarantee data quality, variables with $\geq$60% missing values were discarded.

In our COVID-19 & Air Pollution Working Group, we considered of particular interest to study the influence of socioeconomic and air quality factors on the severity of COVID-19, also motivated by the growing evidence from the literature (Introduction).

Since obvious confidentiality issues prohibited having individualized information regarding his/her socioeconomic status, as an approximation we obtained up to 7 socioeconomic variables describing each patient's postcode of residence: average income level, average age, percentage of the population under 18 and over 65 years old, etc. These public data were obtained from the latest census by the Spanish National Statistical Institute (INE, 2019) [31], and re-interpolated from census districts into postcodes [30].

In addition, for the 8 main air pollutants ($PM_{10}$, $PM_{2.5}$, $O_3$, $NO_2$, $NO$, $NO_X$, $SO_2$, $CO$) we obtained daily measurements as published by the corresponding territorial air quality agencies [32–34]. To estimate the distribution of pollution day-by-day and per postcode, we used Bayesian Generalized Additive Models (BGAMs) [35, 36] with latitude, longitude and elevation [30]. We defined the 'chronic' exposure to air pollution by the levels throughout 2019; whereas 'acute' exposure was considered for the 7 days before each patient's admission. For each time window and pollutant, we computed the 50% and 90% quantiles of day-by-day exposures, hence totalling 32 pollution variables.

### Feature selection: Algorithms

Let us consider a dataset $X$ with $n$ samples and $d$ features. In high-dimensional cases, where $d$ is not much smaller than $n$, it is often convenient to retain just a reduced subgroup of features [37]: to help circumventing the so-called 'curse of dimensionality' (i.e. sparsity of the $n$ data

points in $\mathbb{R}^d$), to simplify the ML models (making them easier to interpret), to shorten their training times, etc.

In practise, dimensionality reduction techniques assume that the input data $X$ contains some features which are either redundant, irrelevant or carry limited information with respect to the outcome of interest $Y$. Hence, it should be possible to remove these features without much loss of information.

FS algorithms incorporate search strategies which aim to find the best subset of features, based on different optimality criteria. Three main categories of FS methods can be distinguished: filters, wrappers and embeddeds [16, 38].

*Filter* algorithms account only for the intrinsic properties of the data, to evaluate the relevance of features, and to remove those with lowest relevance. Filters are conceptually simple, computationally fast and independent of any ML model to be used for a subsequent prediction.

An univariate approach is often used: each of the $d$ features is considered separately, ignoring interdependencies/correlations across features. For example, the mutual information (MI)-based filter ranks variables according to the MI value between them and the target outcome $Y$ [38].

Multivariate filters have also been proposed to incorporate feature interdependencies. The minimum redundancy–maximum relevance (mRMR) algorithm [39], and the fast correlation-based filter (FCBF) [40] aim to find the subgroup of variables which provide the most information about $Y$, with as little redundancy as possible across them; using metrics based on MI to characterize the correlation between variables. In addition, Relief-based algorithms (RBA), such as ReliefF and MultiSURF, rank features considering differences between nearest-neighbor instance pairs [41].

*Wrapper* methods search in the space of all possible feature subsets, evaluating a candidate subset on the basis of its predictive power. To do so, they integrate a 'wrapped'/internal ML model within the algorithm: given a certain candidate subset, the model is trained on it and then tested; thus getting a performance score, which relates to the amount of relevant information carried by the candidate. Considering that the size of the search space ($2^d$) grows exponentially with the number of features, search heuristics are required. Depending on the heuristics employed, two main families of wrappers can be distinguished: deterministic and randomized.

Among the deterministic search algorithms, sequential feature selection (SFS) adds [forward] –or removes [backward]– one feature per step [42]. This greedy choice is based on the performance attained by the internal ML model on the different temporary feature subsets, with/without the candidate feature.

Recursive feature elimination (RFE) [43] is also a deterministic type of wrapper, which consists in discarding features recursively, based on an assessment of importance of single features. This assessment results from the training of the 'wrapped' ML model (e.g. weights of regression coefficients). There also exists a cross-validated version of RFE (RFECV) avoiding the need of pre-specifying the size of the feature subset, which is instead selected attending to the overall ML performance score.

Randomized wrapper algorithms for FS include genetic algorithms (GA) and binary particle swarm optimization (BPSO) as search heuristics. Candidate FS solutions are represented by individuals within a population; whereas a scoring/'fitness' function evaluates their quality. The difference between GA and BPSO lies mainly in the techniques used to 'evolve' from one population to another across generations: GA mimics principles of genetics and natural selection [44], whereas BPSO uses a kind of motion simulating a swarm [45] to go through the search space.

*Embedded* methods lodge, or 'embed', the search for the optimal feature subset within the construction of the ML model: FS is done during the process of ML training [16, 38]. Likewise wrappers, embeddeds are specific to a given learning algorithm. But instead of using the ML models to evaluate each candidate feature subset, they train the ML model just once and then select certain features based on their importance. In this manner, embedded are normally much less computationally intensive than wrappers.

For example, for linear prediction models, when the $L^1$-norm penalty is introduced in the loss function for ML fitting, many of the estimated model coefficients become zero. Thus, FS could consist simply in choosing those features whose coefficients are non-zero.

## Feature selection: Assessment of performance

**Stability.**    The 'stability' of a FS algorithm relates to the reproducibility of its results: if a small change in the dataset $X$ leads to a large change in the subset $S$ of selected features, then the algorithm should be deemed as unstable with respect to data.

To study stability, let us apply the FS algorithm to $X^{(m)}$, the $m$-th out of $M$ different bootstrap samples of our original dataset $X$. The outcome for FS can be summarized in a matrix $\mathcal{Z}$:

$$\mathcal{Z} = \begin{pmatrix} z_{1,1} & z_{1,2} & \cdots & z_{1,d} \\ z_{2,1} & z_{2,2} & \cdots & z_{2,d} \\ \vdots & \vdots & \ddots & \vdots \\ z_{M,1} & z_{M,2} & \cdots & z_{M,d} \end{pmatrix} \tag{1}$$

where $z_{m,i} = 1$ if the $i$-th feature was selected during the $m$-th iteration with the $X^{(m)}$ dataset sample, and $z_{m,i} = 0$ otherwise.

From the matrix $\mathcal{Z}$, stability $\Phi$ is estimated as follows [14]:

$$\hat{\Phi}(\mathcal{Z}) = 1 - \frac{\frac{1}{d}\sum_{i=1}^{d} s_i^2}{\mathbb{E}\left[\frac{1}{d}\sum_{i=1}^{d} s_i^2 | H_0\right]} = 1 - \frac{\frac{1}{d}\sum_{i=1}^{d} s_i^2}{\frac{\bar{n}_{fs}}{d}\left(1 - \frac{\bar{n}_{fs}}{d}\right)} \tag{2}$$

where $\bar{n}_{fs} = \frac{1}{M}\sum_{m=1}^{M}\sum_{i=1}^{d} z_{m,i}$ is the average number of selected features; $H_0$ is the hypothesis standing that for each row of $\mathcal{Z}$, all the subsets of the same size have the same probability of being chosen; $s_i^2 = \frac{M}{M-1}\hat{p}_i(1 - \hat{p}_i)$ is the unbiased sample variance of the selection of the $i$-th feature $X_i$; and $\hat{p}_i = \frac{1}{M}\sum_{m=1}^{M} z_{m,i}$ is the frequency with which the $i$-th feature is chosen.

As the number of bootstrap samples $M$ increases, the estimator $\hat{\Phi}(\mathcal{Z})$ gets closer to the true stability $\Phi$ [14]. However, it is convenient to maintain reasonable computation times, as some FS algorithms tend to be slow.

**Similarity.**    A high 'similarity' between different –possibly stable– FS algorithms may add evidence about the relevance of the selection. As our measure for similarity, here we opted for the Jaccard index. Let $\mathcal{Z}_A^{(m)} = (z_{m,1}^A \ z_{m,2}^A \ \cdots \ z_{m,d}^A)$ and $\mathcal{Z}_B^{(m)} = (z_{m,1}^B \ z_{m,2}^B \ \cdots \ z_{m,d}^B)$ be the selections made by two FS algorithms (A and B) on the same $m$-th bootstrap sample $X^{(m)}$, i.e. the $m$-th rows of matrices $\mathcal{Z}_A$, $\mathcal{Z}_B$. Then the Jaccard index $J$ is defined as the cardinality of the

intersection between both sets, divided by the size of their union:

$$J(\mathcal{Z}_A^{(m)}, \mathcal{Z}_B^{(m)}) = \frac{|\mathcal{Z}_A^{(m)} \cap \mathcal{Z}_B^{(m)}|}{|\mathcal{Z}_A^{(m)} \cup \mathcal{Z}_B^{(m)}|} = \frac{|\mathcal{Z}_A^{(m)} \cap \mathcal{Z}_B^{(m)}|}{|\mathcal{Z}_A^{(m)}| + |\mathcal{Z}_B^{(m)}| - |\mathcal{Z}_A^{(m)} \cap \mathcal{Z}_B^{(m)}|}$$
$$= \frac{\mathcal{Z}_A^{(m)} \cdot \mathcal{Z}_B^{(m)}}{\mathcal{Z}_A^{(m)} \cdot \mathcal{Z}_A^{(m)} + \mathcal{Z}_B^{(m)} \cdot \mathcal{Z}_B^{(m)} - \mathcal{Z}_A^{(m)} \cdot \mathcal{Z}_B^{(m)}}$$

(3)

Consequently, to determine the overall similarity between FS algorithms A and B, we calculated the mean value of the Jaccard index across all bootstrap pairs:

$$J(\mathcal{Z}_A, \mathcal{Z}_B) = \frac{1}{M} \sum_{m=1}^{M} J(\mathcal{Z}_A^{(m)}, \mathcal{Z}_B^{(m)})$$

(4)

**Computation time.**   This manuscript focuses primarily on studying the properties of a range of FS techniques (intra-algorithms' stability, inter-algorithm's similarity), when applied to our motivation COVID-19 dataset for SARS-CoV-2 pneumonia severity. However, a canonical way of exploiting FS would be as part of a ML pipeline with additional stages, including the subsequent training of the ML estimator for severity prediction itself. Therefore, it is desirable for a FS algorithm to be at least relatively fast, to avail for the rest of the pipeline. In other words, given two algorithms which are stable (Stability) and similar enough to each other in terms of results (Similarity), the faster computation may be more practical. For this reason, here we also analysed FS computation times.

## Experimental design

**Data preparation.**   The specificities of our dataset demanded various stages of pre-processing. First of all, the discrete categorical variables (i.e. without intrinsic order, e.g. type of bronchological comorbidity) were transformed into binary features via one-hot encoding [46]; whereas discrete ordinal variables (e.g. qSOFA clinical score) were treated as integer data.

To equalize the range of spanned values across features, with minimal sensitivity to outliers, we opted for a 'robust' scaling procedure: using feature-wise median and inter-quartile range instead of standardization with mean and standard deviation.

Furthermore, missing data were very frequent in our dataset. However, the ample majority of FS algorithms (also, the ML models which constitute the internal part of wrappers and embeddeds) are unable to handle them. Thus, we examined two different imputation strategies, namely: $k$-nearest neighbors imputation ($k$nn) [47] (with $k = 9$ neighbors), and iterative imputation [48] (with $n_{imp} = 4$ features to estimate the missing values). We selected such values for the $k$, $n_{imp}$ hyperparameters via a preliminary exploratory stage, controlling the computational burden of imputation. Of note, large $n_{imp}$ slowed down the iterative imputation to a remarkable extent.

Another key aspect to consider in our motivating COVID-19 dataset was its marked class imbalance, in terms of number of patients by pneumonia severity class. Whenever suitable, we considered balancing the under-represented classes, especially if there was some form of ML model training involved during FS. In the case of filters, we did not apply any balancing, as it is recommended and customary [49]. For embeddeds and wrappers, we used random over-sampling (ROS) [50]. Whenever cross-validated assessments of ML performance were involved (i.e. with wrappers), we implemented our ROS always after the allocation in folds. For embeddeds, where the internal ML training is carried out at once, a mild shrinkage factor (0.01) was introduced with ROS, to add a certain degree of dispersion in the instances.

To prevent any type of information leakage, the full pipeline of pre-processing steps (encoding, scaling, imputation and balancing) was carried out independently for each bootstrap iteration of FS.

**Machine learning goal.** Predicting our target variable $Y$ (SARS-CoV-2 pneumonia severity) may be addressed as an ordinal classification problem [51], as the 3 severity classes entail a clear natural order. Hence, here we considered not only FS approaches for multi-class classification; but also for regression tasks: low, medium and high severities where assigned to 0, 1 and 2 values, respectively [51].

**Set-up of the feature selection algorithms.** Many FS algorithms require to pre-establish $n_{FS}$, the number of desired features to select. In such case, we studied four different choices for $n_{FS}$: 5, 10, 20 and 40 features. Choices for other hyperparameters, which are specific for each FS algorithm, are detailed below. For the interested readership, supplementary materials (S1 Appendix S.C) and Hayet-Otero [52] [Chapter 5] contain detailed explanations and intermediate performance results which supported us in the task of hyperparameter selection.

*Filters.* For the FCBF filter, its $\delta$ hyperparameter controls the threshold to discern between selected and discarded features [40]. Its default value $\delta = 0$ worked acceptably, whereas in a preliminary assessment: $\delta < 0$ yielded noticeable lower stability, and $\delta > 0$ selected extremely few features. In the case of mRMR, we analysed both its MID and MIQ variants (Mutual Information Difference, Quotient) [39]. For ReliefF, its $k$ hyperparameter controls the number of neighbors to assess feature importance. Here we opted for its most widely adopted option: $k = 10$ [41]; as well as for a higher value: $k = 100$, which aims for more accuracy in the importance scores, at the expense of heavier computations.

*Wrappers.* Setting-up an implementation for wrapper-based FS involves notably more aspects to account for than in the case of filters, since one needs to choose aspects related to search heuristics, along with the hyperparameters for the 'wrapped' ML estimation model.

With regard to the ML estimators, here we explored algorithms with different working principles:

1. Linear models, with $L^2$-norm penalization to ensure that the estimator learns without overfitting.

*a*) For the classification approach, $L^2$-norm penalized logistic regression (LR).

*b*) To address the ordinal severity prediction problem as a regression task (Machine Learning goal, [51]), $L^2$-norm penalized regression (i.e. Ridge).

2. Non-linear models:

*I*) $k$-nearest neighbors ($k$NN) [53]: The algorithm carries out its estimation attending to the $k$ instances with lowest distance to the current sample.

*a*) $k$NN classifier ($k$NNC): The label decision is made via neighbors 'voting'.

*b*) $k$NN regressor ($k$NNR): The estimation is an average of the values for the neighbors.

*II*) Histogram-based gradient boosting (HGB): An efficient implementation of the classic Gradient Boosting algorithm [37], it includes the discretization of the continuous features via binning. Notably, HGBs support missing values.

*a*) HGB classifier (HGBC).

*b*) HGB regressor (HGBR).

This choice for the 'wrapped' model was based on a trade-off between: *a*) exploring different families of ML algorithms, *b*) models with as few key hyperparameters as possible (this aiming to ease our task of hyperparameter tuning, and hence to remain robust with respect to such choice; e.g. SVMs would have required to choose the type of kernel, its kernel width and the strength of regularization), *c*) keeping constrained computational demands for model fitting.

For $L^2$-LR, one must determine a suitable value for its regularization penalty parameter *C*. To do so, we conducted preliminary grid and Bayesian cross-validated searches, for all *M* bootstrap samples. Attending to the histogram of optimal choices (S.C, S20:a Fig in S1 Appendix), we set *C* = 0.001. With the same procedure, we chose Ridge's regularization as $\alpha$ = 0.1 (S20:b Fig in S1 Appendix). For KNNC and KNNR, we chose their weight function to be the inverse of the distance between points, whereas the number of neighbors was selected as *k* = 5 by grid search (S20:c Fig in S1 Appendix). For both HGBC and HGBR, our preliminary examinations pointed out that computational demands were high, so we specified a maximum number of 10 learning iterations, as well as 25 histogram bins.

In all of the aforementioned searches for optimal hyperparameter values, as performance goal we used the geometric mean score (GMS) of true positive rates (TPR), i.e. per-class sensitivities [54]:

$$\text{GMS} := \left( \prod_{c=1}^{C} \text{TPR}_{[c]} \right)^{1/C} \tag{5}$$

with *C* = 3 classes here. We opted for this strategy at the view of the remarkable class imbalanced in our problem.

Regarding the wrapper's heuristic search strategy, RFE and RFECV require the 'wrapped' ML model –once fitted– to assess the importance of each feature under consideration. Therefore, just the linear models ($L^2$-LR, Ridge) could be employed, since they are the only ones with such capability.

For some wrappers (SFS, RFE), the number $n_{FS}$ of desired features is fixed *a priori* by design. Whenever it is not (RFECV, GA, BPSO), one should define a fitness function $f(\cdot)$ which reflects the desired objective for FS. On the one hand, one would like the 'wrapped' ML model to be able to achieve accurate estimation results (classification/regression). On the other hand and since we are addressing a FS task, one would logically reward subsets with fewer features. With this in mind, we opted for the following fitness function, based on Vieira *et al.*'s proposal [55]:

$$f(n_{FS}) := \gamma \ score_{ML} + (1 - \gamma)\left(1 - \frac{n_{FS}}{d}\right) \tag{6}$$

with GM from Eq (5) as the $score_{ML}$ quantifying the performance by the 'wrapped' ML model, and $\gamma$ being a weight which controls the trade-off between $score_{ML}$ and subset size ($\gamma$ = 0.8 throughout this work).

The wrappers with a randomized search strategy (GA, BPSO) require several key hyperparameters to be set, concerning details on how to conduct the search across all $2^d$ possible features subsets. Even though the literature provides some guidance on their role and influence, it is advisable to adjust them for each specific application. Here we tried several general-purpose configurations [52], which proved satisfactory convergence of the search given the hyperparameters (further details can be found in the supplementary materials: S.C, S21 Fig in S1 Appendix, as well as in Hayet-Otero [52] [Section 5.3.2]).

In the case of GA, we opted for a population of $n_{pop} = 100$ individuals –to ensure a wide enough space search–, with two mutation probabilities: $p_m = 0.001$ (for a faster convergence), and $p_m = 0.020$ (for a more exploratory algorithm). In both cases, the crossover rate was fixed to an intermediate value of $p_{cx} = 0.70$. Preliminary tests demonstrated that appropriate convergence was achieved with $n_{gen} = 500$ and 1000 generations, respectively.

In order to choose BPSO hyperparameters, we followed a similar approach as for GA. We fixed the inertia term $\omega = 0.9$, and selected two configurations with $|v|_{\max} = 2$ and $|v|_{\max} = 6$: to have wider and narrower exploratory versions, respectively. Other relevant BSPO values were chosen as: $\varphi_p$, $\varphi_g = 0.5$, $n_{pop} = 30$ and $n_{gen} = 2000$ (see S.C, S22 Fig in S1 Appendix, and Hayet-Otero [52] [Section 5.3.2]).

Nevertheless, we had to exclude from our study the scenarios consisting of GA and BPSO with HGB algorithms 'wrapped' inside; since that these particular combinations were extremely slow to compute (longer than one day per iteration); and hence of negligible practical use with FS in a complete ML pipeline.

*Embeddeds*. For the embedded methods, the same type of linear ML estimators as for wrappers were employed. Nevertheless, in this case with $L^1$-norm regularization instead of $L^2$: i.e. $L^1$-norm penalized LR classification, and Lasso (instead of Ridge) for the regression approach to ordinal classification. Controlling the respective regularization term ($C$ for $L^1$-LR, $\alpha$ for Lasso), different degrees of dimensionality reduction could be achieved. In particular: for $L^1$-LR we explored $C = \{0.075, 0.050, 0.025, 0.010, 0.005\}$, whereas for Lasso $\alpha = \{0.005, 0.010, 0.025, 0.050, 0.075\}$. Both sets are listed in order from lesser to more regularization.

**FS performance evaluation.** To characterize the behaviour of the different FS presented in Feature Selection: Algorithms on our data, we studied their performance in terms of stability, similarity and computational complexity, as introduced in Feature Selection: Assessment of performance.

For an exhaustive analysis, each FS algorithm was run using $M = 100$ bootstrap samples $X^{(m)}$ of the original dataset $X$. The random seed for bootstrap sampling was controlled, to guarantee that the sequences of $X^{(m)}$ samples were the same for all FS algorithms; hence availing for comparable simulations and results, when calculating Jaccard similarities with Eq (4).

**Implementation details.** The code for our analyses (Code sharing) was implemented in Python, based on publicly available libraries for ML, FS and optimization. Most of the modules used `scikit-learn` [56]: including for pre-processing (encoding, scaling, imputation), for the internal ('wrapped'/'embeded') ML estimators, as well as for the MI filters, deterministic wrappers and embedded methods in FS. Resampling for class balancing was performed via `imbalanced-learn` [57], which also provided the GMS score in Eq (5). Other FS algorithms used dedicated libraries: mRMR filters from `PymRMR` [58], FCBF from `scikit-feature` [59], RBAs from `scikit-rebate` [60], as well as heuristic optimization libraries—GA from `feature-selection-ga` [61] and BPSO from `PySwarms` [62]. Bayesian hyperparameter tuning was implemented with `scikit-optimize` [63]. Code parallelization for efficient computation was carried out by means of `pathos` library [64], whereas analyses on FS stability were conducted using the code provided by Nogueira *et al.* [14] in their GitHub repository. Graphic visualizations were produced with `matplotlib` [65], `seaborn` [66], `statsmodels` [67], `PtitPrince` [68], and `geopandas` [69].

All experiments were run on a computer cluster at the Basque Center for Applied Mathematics (BCAM). Regarding computation times, these were measured for the specific part of code corresponding to the FS algorithms themselves, i.e. without the remaining ML pipeline (pre-processing, etc.).

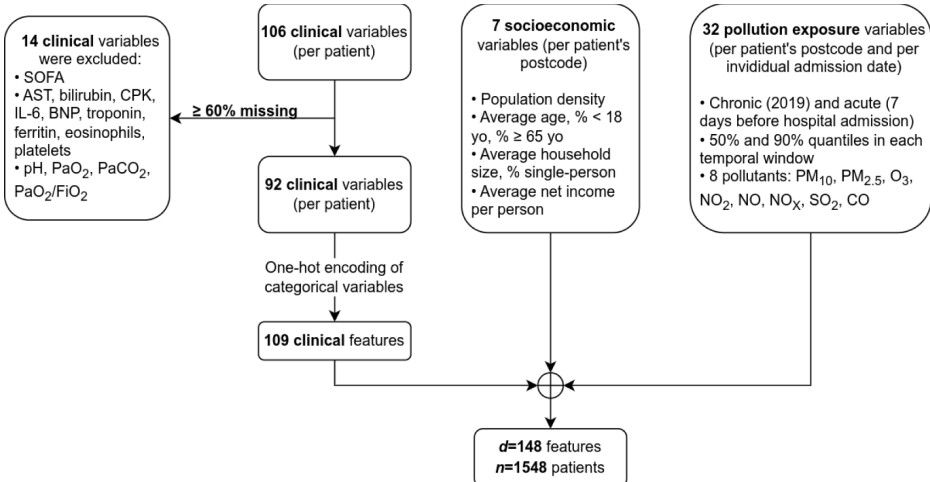

**Fig 1. Flow chart for the included and excluded variables, and feature encoding.**

## Results

### Data collection

Our study enrolled a cohort of $n$ = 1548 patients. Up to total of 106 different clinical variables –attributes– were recorded for each patient ([Fig 1]). Based on our data quality-assurance criterion, we had to discard 14 variables with ⩾60% missing values. These were: the Sequential Organ Failure Assessment score (SOFA); 9 biomarkers from blood tests—aspartate aminotransferase (AST, a.k.a. SGOT), bilirubin, creatine phosphokinase (CPK), interleukin-6 (IL-6), brain natriuretic peptide (BNP), troponin, ferritin, eosinophils, and platelets; as well as 4 results from arterial blood gas tests—acid-base balance (pH), partial pressures of $O_2$ and $CO_2$ ($PaO_2$, $PaCO_2$), and the $PaO_2/FiO_2$ ratio. The remaining 92 variables became 109 features, after one-hot encoding of categorical variables into binary (see the Data preparation section). Together with 7 socioeconomic factors and 32 exposures to air pollution, our dataset was hence composed of $d$ = 148 features. Supplementary materials (S1 Appendix S.A) contains a detailed list of them.

S2 Table in S1 Appendix, alongside S1–S19 Figs (S1 Appendix S.B), provide a detailed characterization of the clinical, socioeconomic and pollution exposure data [70] for our cohort; both in overall and grouped by SARS-CoV-2 pneumonia severity. In the aforementioned S1 Appendix, univariate statistical comparisons for discrete variables were performed by means of the $\chi^2$ test, whereas its effect size was calculated with the bias-corrected Cramer's $V$ [71]. For continuous variables, univariate comparisons were made with the non-parametric Kruskal-Wallis test, and its corresponding $\eta_H^2$ effect size. Thresholds for interpreting effect sizes were taken as recommended by Cohen [72].

For the sake of compactness, Table 1 contains the distribution in terms of number of cases: total, by SARS-CoV-2 pneumonia severity group, and by hospital (anonymized).

### Feature selection: Stability

Tables 2 to 4 contain the mean estimated stability $\hat{\Phi}$ for each FS scenario, along with its 95% confidence interval (CI), calculated via bootstrapping as proposed by Nogueira *et al.* [14].

**Table 1. Distribution of patients in our cohort.** Total number and percentage of cases: by SARS-CoV-2 pneumonia severity group, and by hospital (anonymized).

| Hospital | Total | By severity | | |
|---|---|---|---|---|
| | | Low | Medium | High |
| | n = 1548 | n = 712 (46.0%) | n = 238 (15.4%) | n = 598 (38.6%) |
| A | 358 (23.1%) | 205 (57.3%) | 36 (10.1%) | 117 (32.7%) |
| B | 380 (24.5%) | 229 (60.3%) | 50 (13.2%) | 101 (26.6%) |
| C | 438 (28.3%) | 119 (27.2%) | 59 (13.5%) | 260 (59.4%) |
| D | 372 (24.0%) | 159 (42.7%) | 93 (25.0%) | 120 (32.3%) |

Similar information regarding the different algorithms' computation times can be found in the on-line supplementary materials (S1 Appendix S.E).

Although the roles of the imputer and the subsequent FS algorithm itself are difficult to separate, given that our dataset $X$ contains a large portion of missing values in many features, the type of imputation influenced notably the stability of the overall FS pipeline. The iterative imputer (column-based, i.e. feature-based), as opposed to the $k$nn imputer (which is basically row-/instance-based), could be introducing an extra amount of correlation between features, hence adding difficulty to the FS task.

**Table 2. Stability for the filter algorithms.** Mean and 95% CI.

| Imputer | FS: Filter | | Number of features to select | | | | |
|---|---|---|---|---|---|---|---|
| | | | $n_{FS} = 5$ | $n_{FS} = 10$ | $n_{FS} = 20$ | $n_{FS} = 40$ | Not pre-fixed |
| $k$nn | MI | Classif. | 0.3896 | 0.5305 | 0.7286 | 0.8851 | — |
| | | | [0.3614, 0.4178] | [0.5088, 0.5521] | [0.7138, 0.7434] | [0.8773, 0.8929] | |
| | | Regress. | 0.3781 | 0.5054 | 0.7294 | 0.8734 | — |
| | | | [0.3499, 0.4063] | [0.4852, 0.5255] | [0.7156, 0.7432] | [0.8641, 0.8826] | |
| | mRMR | MID | 0.4610 | 0.4609 | 0.5646 | 0.6469 | — |
| | | | [0.4230, 0.4990] | [0.4426, 0.4793] | [0.5482, 0.5811] | [0.6341, 0.6597] | |
| | | MIQ | 0.4357 | 0.4557 | 0.5192 | 0.6502 | — |
| | | | [0.4056, 0.4657] | [0.4321, 0.4793] | [0.5049, 0.5335] | [0.6380, 0.6623] | |
| | FCBF | $\delta = 0$ | — | — | — | — | 0.3520 |
| | | | | | | | [0.3263, 0.3777] |
| Iterat. | MI | Classif. | 0.4836 | 0.6471 | 0.9171 | 0.7573 | — |
| | | | [0.4516, 0.5155] | [0.6284, 0.6657] | [0.9097, 0.9245] | [0.7496, 0.7650] | |
| | | Regress. | 0.4874 | 0.6442 | 0.9093 | 0.7385 | — |
| | | | [0.4577, 0.5170] | [0.6260, 0.6625] | [0.9007, 0.9179] | [0.7309, 0.7460] | |
| | mRMR | MID | 0.5913 | 0.5517 | 0.6300 | 0.6096 | — |
| | | | [0.5563, 0.6264] | [0.5281, 0.5753] | [0.6115, 0.6485] | [0.5988, 0.6204] | |
| | | MIQ | 0.5787 | 0.5352 | 0.5419 | 0.6137 | — |
| | | | [0.5303, 0.6270] | [0.5086, 0.5617] | [0.5255, 0.5583] | [0.6026, 0.6247] | |
| | FCBF | $\delta = 0$ | — | — | — | — | 0.4759 |
| | | | | | | | [0.4451, 0.5067] |
| None | ReliefF | $k = 10$ | 0.5296 | 0.5639 | 0.6342 | 0.6616 | — |
| | | | [0.4953, 0.5639] | [0.5404, 0.5875] | [0.6145, 0.6538] | [0.6490, 0.6741] | |
| | | $k = 100$ | 0.7519 | 0.8292 | 0.7258 | 0.7425 | — |
| | | | [0.7267, 0.7771] | [0.8147, 0.8437] | [0.7129, 0.7387] | [0.7310, 0.7540] | |
| | MultiSURF | | 0.7874 | 0.8175 | 0.7635 | 0.7288 | — |
| | | | [0.7621, 0.8127] | [0.8001, 0.8349] | [0.7488, 0.7782] | [0.7164, 0.7412] | |

**Table 3. Stability for the wrapper algorithms.** Mean and 95% CI.

| Imputer | FS: Wrapper | | | Number of features to select | | | | |
|---|---|---|---|---|---|---|---|---|
| | Search | Params | ML | $n_{FS} = 5$ | $n_{FS} = 10$ | $n_{FS} = 20$ | $n_{FS} = 40$ | Not pre-fixed |
| $k$nn | SFS | Forward | $L^2$-LR | 0.2994 [0.2693, 0.3294] | 0.1980 [0.1800, 0.2161] | 0.1576 [0.1460, 0.1692] | 0.2260 [0.2134, 0.2387] | — |
| | | | Ridge | 0.4289 [0.3945, 0.4633] | 0.2983 [0.2762, 0.3204] | 0.1892 [0.1776, 0.2008] | 0.1484 [0.1384, 0.1584] | — |
| | | | $k$NNC | 0.0992 [0.0821, 0.1163] | 0.1291 [0.1138, 0.1445] | 0.2756 [0.2574, 0.2938] | 0.4914 [0.4747, 0.5081] | — |
| | | | $k$NNR | 0.1471 [0.1225, 0.1717] | 0.1507 [0.1357, 0.1657] | 0.3199 [0.3018, 0.3381] | 0.5060 [0.4874, 0.5246] | — |
| | RFE | — | $L^2$-LR | 0.7560 [0.7189, 0.7935] | 0.6641 [0.6489, 0.6793] | 0.7292 [0.7144, 0.7441] | 0.6472 [0.6363, 0.6582] | — |
| | | | Ridge | 0.4371 [0.1014, 0.4728] | 0.4486 [0.4148, 0.4824] | 0.3962 [0.3764, 0.4161] | 0.3871 [0.3711, 0.4031] | — |
| | RFECV | — | $L^2$-LR | — | — | — | — | 0.5627 [0.5362, 0.5893] |
| | | | Ridge | — | — | — | — | 0.3604 [0.3458, 0.3750] |
| | GA | $p_m = 0.001$ | $L^2$-LR | — | — | — | — | 0.0743 [0.0663, 0.0824] |
| | | | Ridge | — | — | — | — | 0.0824 [0.0735, 0.0912] |
| | | | $k$NNC | — | — | — | — | 0.0322 [0.0265, 0.0491] |
| | | | $k$NNR | — | — | — | — | 0.0289 [0.0234, 0.0343] |
| | | $p_m = 0.020$ | $L^2$-LR | — | — | — | — | 0.0949 [0.0860, 0.1038] |
| | | | Ridge | — | — | — | — | 0.0977 [0.0882, 0.1072] |
| | | | $k$NNC | — | — | — | — | 0.0428 [0.0365, 0.0491] |
| | | | $k$NNR | — | — | — | — | 0.0485 [0.0416, 0.0553] |
| | BPSO | $\omega = 2$ | $L^2$-LR | — | — | — | — | 0.0341 [0.0280, 0.0402] |
| | | | Ridge | — | — | — | — | 0.0524 [0.0454, 0.0594] |
| | | | $k$NNC | — | — | — | — | 0.0161 [0.0112, 0.0209] |
| | | | $k$NNR | — | — | — | — | 0.0193 [0.0140, 0.0246] |
| | | $\omega = 6$ | $L^2$-LR | — | — | — | — | 0.0389 [0.0326, 0.0452] |
| | | | Ridge | — | — | — | — | 0.0536 [0.0467, 0.0605] |
| | | | $k$NNC | — | — | — | — | 0.0169 [0.0119, 0.0220] |
| | | | $k$NNR | — | — | — | — | 0.0137 [0.0089, 0.0185] |
| Iterat. | SFS | Forward | $L^2$-LR | 0.3274 [0.2947, 0.3600] | 0.2061 [0.1885, 0.2236] | 0.1585 [0.1477, 0.1693] | 0.2351 [0.2213, 0.2489] | — |
| | | | Ridge | 0.4198 [0.3810, 0.4585] | 0.3152 [0.2920, 0.3384] | 0.2003 [0.1857, 0.2150] | 0.1528 [0.1418, 0.1637] | — |
| | | | $k$NNC | 0.1474 [0.1232, 0.1715] | 0.1776 [0.1592, 0.1960] | 0.3020 [0.2814, 0.3226] | 0.4557 [0.4360, 0.4754] | — |
| | | | $k$NNR | 0.2049 [0.1851, 0.2246] | 0.2161 [0.1990, 0.2333] | 0.3803 [0.3604, 0.4001] | 0.5166 [0.4990, 0.5341] | — |
| | RFE | — | $L^2$-LR | 0.4987 [0.4728, 0.5247] | 0.5511 [0.5319, 0.5704] | 0.6497 [0.6336, 0.6658] | 0.6323 [0.6199, 0.6448] | — |
| | | | Ridge | 0.5225 [0.4881, 0.5569] | 0.3895 [0.3658, 0.4132] | 0.3401 [0.3218, 0.3585] | 0.3376 [0.3220, 0.3531] | — |
| | RFECV | — | $L^2$-LR | — | — | — | — | 0.4513 [0.4198, 0.4828] |
| | | | Ridge | — | — | — | — | 0.3010 [0.2872, 0.3147] |
| | GA | $p_m = 0.001$ | $L^2$-LR | — | — | — | — | 0.0795 [0.0716, 0.0874] |
| | | | Ridge | — | — | — | — | 0.0871 [0.0786, 0.0957] |
| | | | $k$NNC | — | — | — | — | 0.0784 [0.0708, 0.0860] |
| | | | $k$NNR | — | — | — | — | 0.0900 [0.0828, 0.0972] |
| | | $p_m = 0.020$ | $L^2$-LR | — | — | — | — | 0.0999 [0.0911, 0.1086] |
| | | | Ridge | — | — | — | — | 0.1159 [0.1062, 0.1257] |
| | | | $k$NNC | — | — | — | — | 0.0978 [0.0901, 0.1056] |
| | | | $k$NNR | — | — | — | — | 0.1217 [0.1142, 0.1292] |
| | BPSO | $\omega = 2$ | $L^2$-LR | — | — | — | — | 0.0428 [0.0370, 0.0486] |
| | | | Ridge | — | — | — | — | 0.0600 [0.0526, 0.0673] |
| | | | $k$NNC | — | — | — | — | 0.0621 [0.0562, 0.0681] |
| | | | $k$NNR | — | — | — | — | 0.0907 [0.0835, 0.0980] |
| | | $\omega = 6$ | $L^2$-LR | — | — | — | — | 0.0435 [0.0370, 0.0499] |
| | | | Ridge | — | — | — | — | 0.0635 [0.0572, 0.0698] |
| | | | $k$NNC | — | — | — | — | 0.0641 [0.0578, 0.0703] |
| | | | $k$NNR | — | — | — | — | 0.0957 [0.0888, 0.1026] |

(*Continued*)

**Table 3.** (Continued)

| Imputer | FS: Wrapper | | | Number of features to select | | | | |
|---|---|---|---|---|---|---|---|---|
| | Search | Params | ML | $n_{FS} = 5$ | $n_{FS} = 10$ | $n_{FS} = 20$ | $n_{FS} = 40$ | Not pre-fixed |
| None | SFS | Forward | HGBC | 0.1639 [0.1382, 0.1897] | 0.1354 [0.1193, 0.1516] | 0.1743 [0.1584, 0.1901] | 0.2670 [0.2479, 0.2860] | — |
| | | | HGBR | 0.2314 [0.2064, 0.2564] | 0.1598 [0.1421, 0.1776] | 0.1187 [0.1087, 0.1286] | 0.1177 [0.1070, 0.1284] | — |
| | GA | $p_m$ | HGBC | — | — | — | — | Exceeds max. runtime* |
| | | | HGBR | — | — | — | — | Exceeds max. runtime* |
| | BPSO | $\omega$ | HGBC | — | — | — | — | Exceeds max. runtime* |
| | | | HGBR | — | — | — | — | Exceeds max. runtime* |

*Computations for GA and BPSO with 'wrapped' HGB estimators were discarded, as they exceeded the maximum runtime of 1 day per bootstrap sample.

For those scenarios where $n_{FS}$ was required to be fixed *a priori*, we may interpret a decay in stability $\hat{\Phi}$ as a choice deviating from the 'optimal' (unknown) number of relevant features. By going beyond that optimum, we would be forcing the FS to include less informative features into the selected subset, thus reducing the overall robustness in terms of stability. However, the observed behaviour is not always concave with $n_{FS}$, and $\hat{\Phi}$ peaks around different $n_{FS}$ values (also with the added effect of the imputer, as mentioned above).

We also evaluated scenarios where $n_{FS}$ was not pre-fixed. Table 5 contains the median and 95% percentile range for $n_{FS}$ in such scenarios. In general, embedded FS schemes with regularization were the more stable the stronger the regularization term was (i.e. smaller $C$ in $L^1$-LR, larger $\alpha$ in Lasso): selecting fewer features. On the other hand, RFECV, GA and BPSO suffered from low stabilities, arguably due to the fact that (except RFECV with $L^2$-LR), the others

**Table 4. Stability for the embedded algorithms.** Mean and 95% CI.

| Imputer | FS: Embedded | | Num. feats: Not pre-fixed |
|---|---|---|---|
| $k$nn | $L^1$-LR | $C = 0.075$ | 0.4923 [0.4799, 0.5046] |
| | | $C = 0.050$ | 0.5252 [0.5137, 0.5368] |
| | | $C = 0.025$ | 0.5704 [0.5589, 0.5820] |
| | | $C = 0.010$ | 0.6400 [0.6264, 0.6537] |
| | | $C = 0.005$ | 0.7647 [0.7446, 0.7849] |
| | Lasso | $\alpha = 0.005$ | 0.3893 [0.3788, 0.3998] |
| | | $\alpha = 0.010$ | 0.5158 [0.5038, 0.5279] |
| | | $\alpha = 0.025$ | 0.6071 [0.5958, 0.6185] |
| | | $\alpha = 0.050$ | 0.7200 [0.7045, 0.7356] |
| | | $\alpha = 0.075$ | 0.7604 [0.7445, 0.7762] |
| Iterat. | $L^1$-LR | $C = 0.075$ | 0.4796 [0.4682, 0.4911] |
| | | $C = 0.050$ | 0.5114 [0.5007, 0.5221] |
| | | $C = 0.025$ | 0.5597 [0.5470, 0.5724] |
| | | $C = 0.010$ | 0.5920 [0.5785, 0.6056] |
| | | $C = 0.005$ | 0.7140 [0.6875, 0.7405] |
| | Lasso | $\alpha = 0.005$ | 0.3722 [0.3623, 0.3822] |
| | | $\alpha = 0.010$ | 0.4941 [0.4814, 0.5067] |
| | | $\alpha = 0.025$ | 0.5865 [0.5758, 0.5972] |
| | | $\alpha = 0.050$ | 0.6735 [0.6594, 0.6875] |
| | | $\alpha = 0.075$ | 0.6783 [0.6621, 0.6944] |

**Table 5. Number of selected features for algorithms with non-fixed $n_{FS}$.** Median and 95% percentile range.

| | | | *knn* imputer | Iterat. imputer |
|---|---|---|---|---|
| **FS: Filter** | | | | |
| Filter | Params | | | |
| FCBF | $\delta = 0$ | | 8.0 [6.0, 10.0] | 5.0 [7.0, 11.0] |
| **FS: Wrapper** | | | | |
| Search | Params | ML | | |
| RFECV | — | $L^2$-LR | 11.5 [3.0, 25.0] | 10.0 [3.0, 22.0] |
| | | Ridge | 46.0 [28.0, 70.2] | 46.0 [23.4, 70.0] |
| GA | $p_m = 0.001$ | $L^2$-LR | 43.0 [32.0, 55.5] | 43.0 [36.0, 55.0] |
| | | Ridge | 52.0 [43.0, 60.6] | 50.5 [41.0, 61.0] |
| | | $k$NNC | 41.0 [31.0, 50.0] | 47.0 [40.0, 55.0] |
| | | $k$NNR | 41.0 [32.0, 48.5] | 48.0 [39.0, 58.0] |
| | $p_m = 0.020$ | $L^2$-LR | 38.0 [31.0, 46.5] | 38.0 [32.0, 45.5] |
| | | Ridge | 48.0 [39.0, 56.0] | 47.0 [40.0, 54.5] |
| | | $k$NNC | 37.0 [30.0, 46.5] | 42.0 [33.5, 49.0] |
| | | $k$NNR | 36.0 [29.0, 45.0] | 41.0 [32.0, 49.5] |
| BPSO | $w = 2$ | $L^2$-LR | 53.0 [46.0, 61.0] | 52.5 [44.0, 62.5] |
| | | Ridge | 60.0 [51.0, 68.5] | 60.0 [51.0, 68.0] |
| | | $k$NNC | 51.0 [43.0, 63.5] | 53.0 [44.5, 63.5] |
| | | $k$NNR | 50.0 [42.5, 58.0] | 53.0 [45.0, 61.5] |
| | $w = 6$ | $L^2$-LR | 52.5 [44.5, 59.5] | 53.0 [40.4, 59.0] |
| | | Ridge | 60.0 [51.0, 66.0] | 60.0 [50.5, 67.5] |
| | | $k$NNC | 52.0 [45.0, 59.0] | 53.0 [44.0, 61.5] |
| | | $k$NNR | 50.0 [42.0, 58.5] | 53.0 [46.5, 62.0] |
| **FS: Embedded** | | | | |
| ML | Params | | | |
| $L^1$-LR | $C = 0.075$ | | 75.0 [69.0, 83.0] | 79.0 [70.5, 87.0] |
| | $C = 0.050$ | | 59.0 [51.5, 65.0] | 63.5 [54.5, 69.0] |
| | $C = 0.025$ | | 34.0 [28.5, 40.0] | 39.0 [32.0, 45.0] |
| | $C = 0.010$ | | 13.0 [10.0, 16.0] | 17.0 [13.0, 21.0] |
| | $C = 0.005$ | | 4.0 [3.0, 7.0] | 5.0 [3.0, 7.0] |
| Lasso | $\alpha = 0.005$ | | 73.0 [66.5, 83.0] | 77.5 [71.0, 86.5] |
| | $\alpha = 0.010$ | | 51.0 [43.5, 57.0] | 54.0 [47.5, 61.0] |
| | $\alpha = 0.025$ | | 28.0 [23.0, 33.0] | 31.0 [25.5, 37.0] |
| | $\alpha = 0.050$ | | 16.0 [13.0, 19.5] | 19.0 [16.0, 24.0] |
| | $\alpha = 0.075$ | | 11.0 [8.5, 14.0] | 13.0 [10.0, 17.0] |

tended to select a large number of variables. This behaviour may be pointing out the intrinsic difficulty of FS with our dataset.

Regarding computational loads (S1 Appendix S.E), wrappers were the most demanding by orders of magnitude: particularly with HGB algorithms as internal ML estimators, and/or with randomized search heuristics (GA, BPSO). Adding to their poor stability, most wrappers appeared to be an impractical choice for FS in our scenario.

In view of stability and computation time, the algorithms which showed the best overall properties were: among the filters, MI-based (univariate) with $n_{FS} = 20$ or $40$, as well as ReliefF (multivariate) with $k = 100$ neighbours. Among wrappers, the RFE with $L^2$-LR with at most

$n_{FS}$ = 20 showed reasonable performance. Regarding embeddeds, both $L^1$-LR and Lasso, with sufficient regularization strengths, were stable and fast.

### Feature selection: Similarity

In addition to studying the stability properties of each FS scenario and its computation time, here we analyzed how much the selected subsets of features resemble each other. Instead of exploring all possible comparisons, we focused on those cases with satisfactory stability as to consider FS results meaningful.

The literature suggests that $\hat{\Phi} \geqslant 0.75$ may indicate high stability [73], whereas values below 0.4 should be rendered as unsatisfactory. Nevertheless, for this work we decided to use a slightly lower cut-off: $\hat{\Phi} \geqslant 0.70$. Merely 21 configurations made it past this threshold: 16 filters, 2 wrappers, and 3 embeddeds.

On the other hand, a comparison between configurations where $n_{FS}$ is set differently may be unfair. Therefore, we analysed algorithms with the same $n_{FS}$ against each other; aside from embeddeds, for which by definition $n_{FS}$ was not pre-established. Average Jaccard similarity results are displayed in Fig 2.

Fig 2 reflects that FS algorithms with comparable configuration tended to result in high Jaccard similarity scores. With the same imputer, the MI-based univariate filters performed similarly, regardless of the type of definition of MI: either for classification or for regression approaches. ReliefF with $k$ = 100 neighbors and MultiSURF yielded also similar results, which could (up to certain extent) be expected, since MultiSURF is an extension of ReliefF with an automatic tuning for $k$. Embedded FS with Lasso regression yielded moderate correspondence between regularization strengths of $\alpha$ = 0.050 and 0.075.

### Feature selection: Selected features

In general, the subsets of features output by FS algorithms of different nature did not tend to coincide highly in our dataset. Nevertheless, we carried out an in-depth examination about which specific features were picked with highest frequencies by each method. Note that pairs of selection sets, with even a moderate-to-low Jaccard similarity, may still consistently agree on a few features, which could thus be deemed as relevant.

For each of the 21 stable algorithms, we focused on those features selected in at least 80% of the $M$ = 100 bootstrap samples. Figs 3–5 depict them, along with their corresponding frequencies. For the sake of clarity, features have been labeled with numbers. Their corresponding names are listed in the on-line supplementary materials (S1 Appendix S.A).

Table 6 contains a ranking of the 20 most selected features, with the top-8 of those having been chosen by more than a half of the stable FS configurations. Thus, there exists certain agreement among the algorithms in highlighting:

*a*) blood levels of C-reactive protein (CRP) [feature #78, ranked 1st];

*b*) PSI score (pneumonia severity index) [#31, ranked 2nd];

*c*) respiratory rate (RR) [#54, ranked 6th] and magnitudes related to oxygenation levels, like: oxygen saturation Sp O2 (measured by a pulse oximeter) [#56, 4th], its quotient Sp O2/RR with respiratory rate [#59, 3th], or the Sat O2/Fi O2 ratio from blood gas tests [feature #92, ranked 8th];

*d*) the neutrophil-to-lymphocyte ratio (NLR) [#86, ranked 5th] –and to some extent, also each of the two cell counts separately [#83 and #82, ranked 12th and 9th]–;

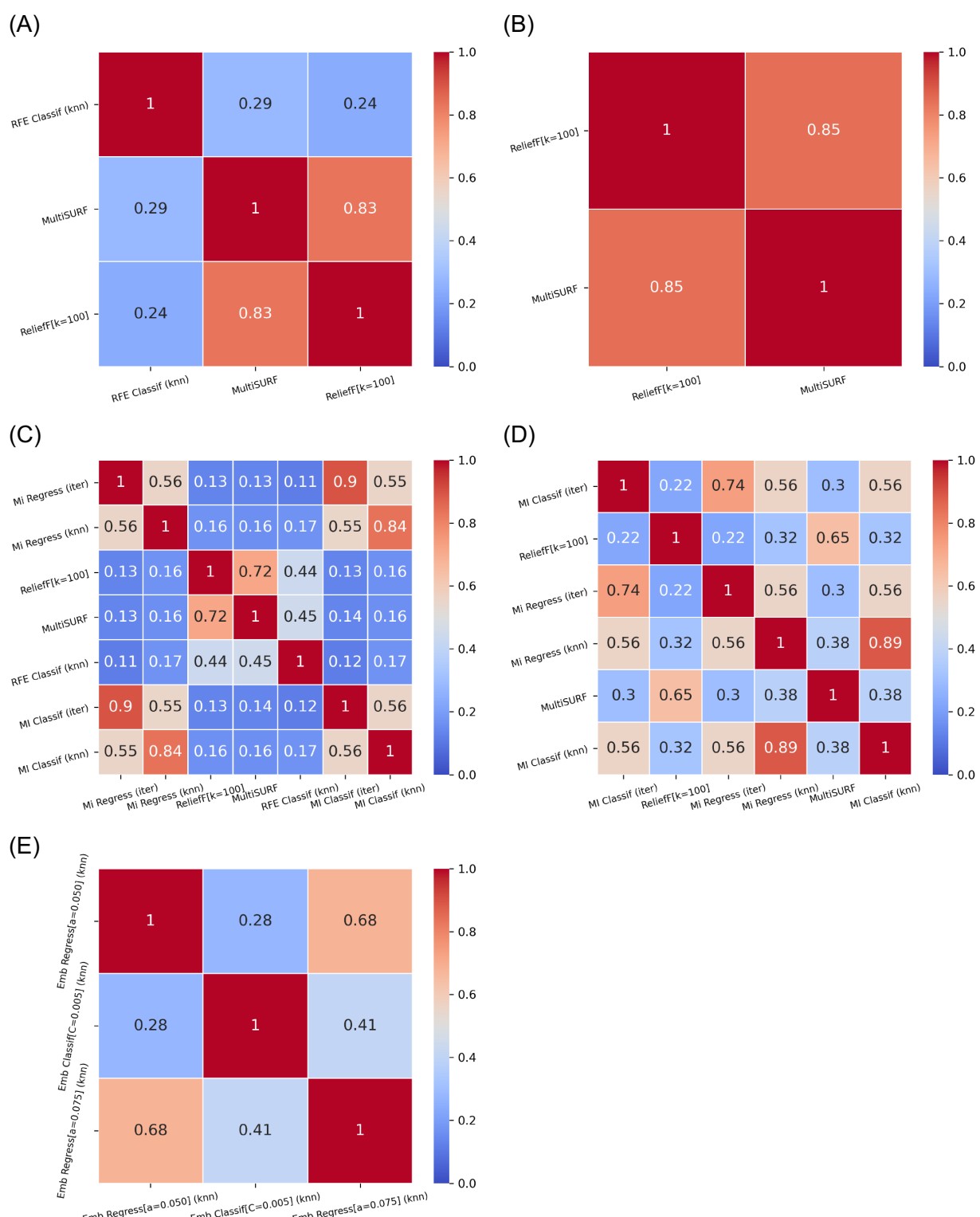

**Fig 2. Jaccard similarity index between feature subsets.** For all pairs of stable algorithms, these grouped by $n_{FS}$ specification. Results were averaged over $M = 100$ bootstrap samples. (**a**) $n_{FS} = 5$. (**b**) $n_{FS} = 10$. (**c**) $n_{FS} = 20$. (**d**) $n_{FS} = 40$. (**e**) $n_{FS}$ not pre-fixed.

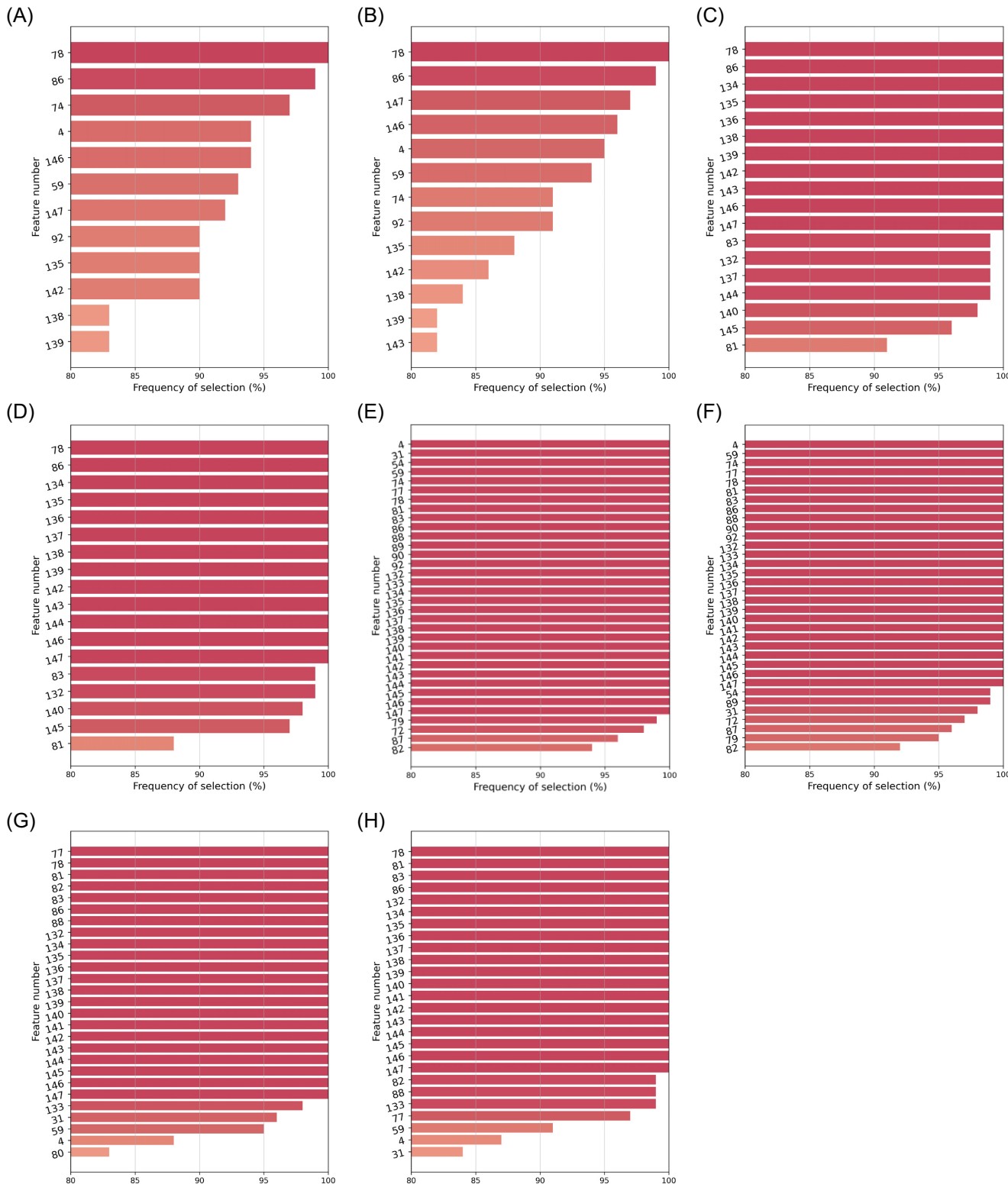

**Fig 3. Features selected in ⩾80% cases by the stable MI filters:** $n_{FS}$ = 20 or 40. (**a**) MI Classif—knn imputer: $n_{FS}$ = 20. (**b**) MI Regress—knn imputer: $n_{FS}$ = 20. (**c**) MI Classif—iterat imputer: $n_{FS}$ = 20. (**d**) MI Regress—iterat imputer: $n_{FS}$ = 20. (**e**) MI Classif—knn imputer: $n_{FS}$ = 40. (**f**) MI Regress—knn imputer: $n_{FS}$ = 40. (**g**) MI Classif—iterat imputer: $n_{FS}$ = 40. (**h**) MI Regress—iterat imputer: $n_{FS}$ = 40.

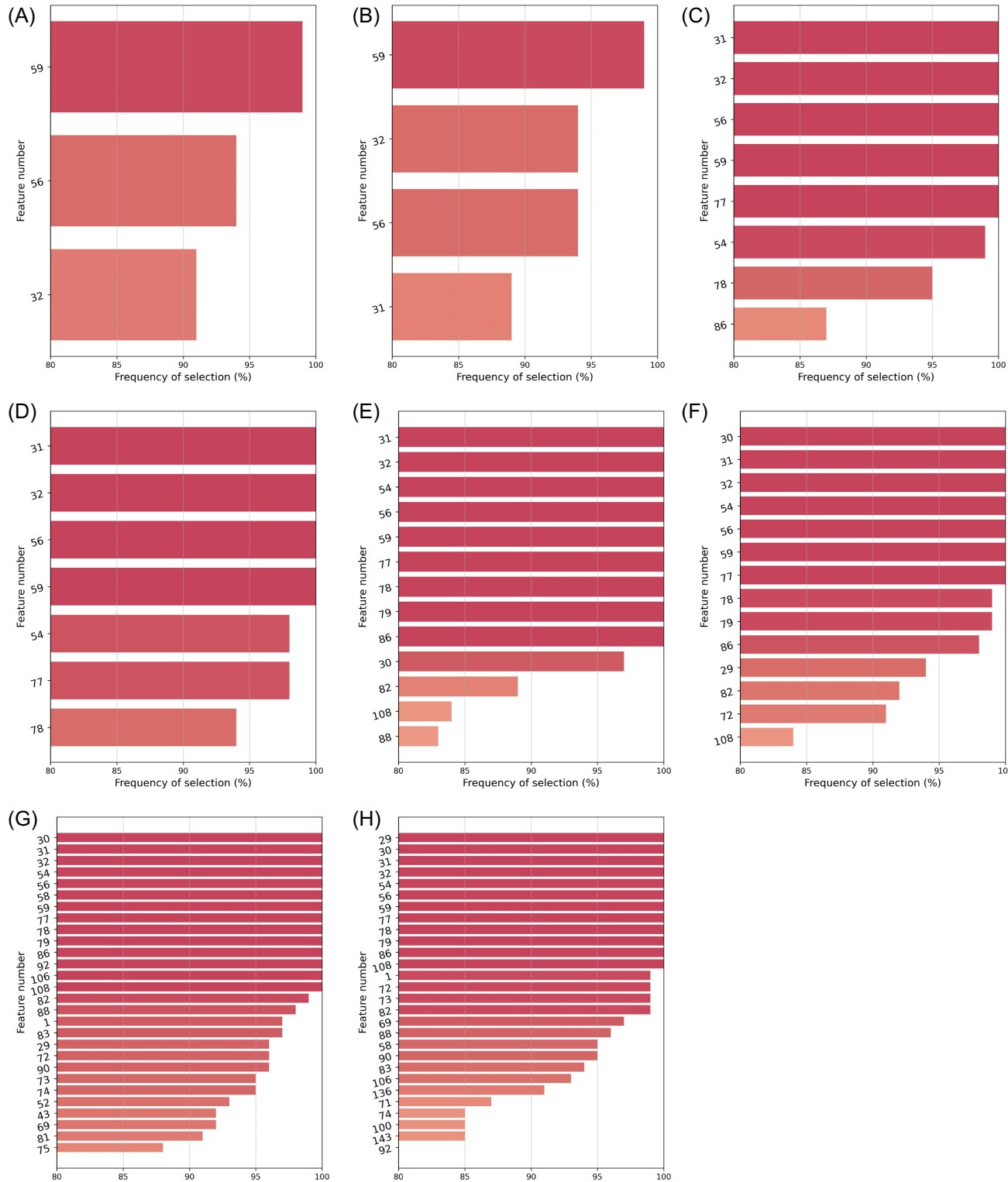

**Fig 4. Features selected in ⩾80% cases by the stable RBA filters: All of them without imputation.** (a) ReliefF ($k$ = 1 00): $n_{FS}$ = 5. (b) MultiSURF: $n_{FS}$ = 5. (c) ReliefF ($k$ = 100): $n_{FS}$ = 10. (d) MultiSURF: $n_{FS}$ = 10. (e) ReliefF ($k$ = 100): $n_{FS}$ = 20. (f) MultiSURF; $n_{FS}$ = 20. (g) ReliefF ($k$ = 100): $n_{FS}$ = 40. (h) MultiSURF: $n_{FS}$ = 40.

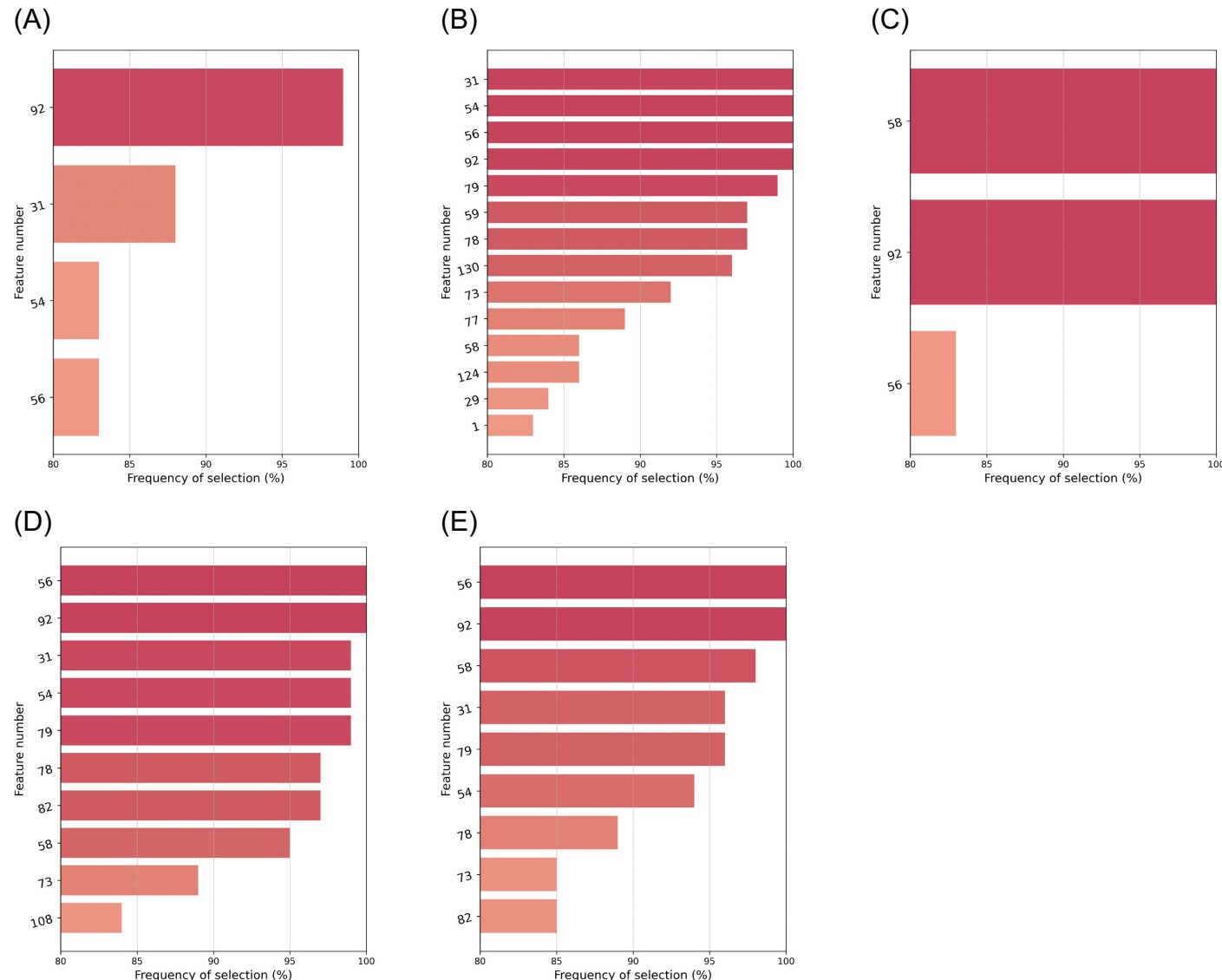

**Fig 5. Features selected in ⩾80% cases by the stable RFE wrappers (*a,b*) and embeddeds (*c–e*): All of them with the *k*nn imputer.** (**a**) RFE: $n_{FS} = 5$. (**b**) RFE: $n_{FS} = 20$. (**c**) $L^1$-LR: C = 0.005. (**d**) Lasso: $\alpha = 0.050$. (**e**) Lasso: $\alpha = 0.075$.

*e*) lactate dehydrogenase (LDH) [#77, 7th]; and

*f*) procalcitonin (PCT) levels [#79, 10th] in blood.

In spite of lower degrees of agreement across all configurations, some features were consistently chosen by the same type of FS algorithm: e.g. lymphocites [#82] and PCT [feature #79] were relevant for 2 out of 3 embeddeds (as well as for some filters). All 8 RBAs picked the qSOFA score for sepsis [#32], whereas 6 out of 8 MI algorithms and a few RBAs emphasized neutrophils [#83] and leukocytes [#81]. Notably, univariate MI-based filters found acute levels of air pollution by $NO_2$ [#135, #143], $SO_2$ [#138, #146], CO [#139, #147] and $O_3$ [#142], to be relevant for the clinical outcome of SARS-CoV-2 pneumonia severity.

**Table 6. Top-20 selected features.** Out of the 21 stable FS configurations, how many of them selected a certain feature for ⩾80% of the $M = 100$ bootstrap iterations.

| Rank | Feat. num. | Filters | | Wrappers | Embedded | Total | % |
|---|---|---|---|---|---|---|---|
| | | MI | RBA | RFE | | | |
| | | *(Max. 8)* | *(Max. 8)* | *(Max. 2)* | *(Max. 3)* | *(Max. 21)* | |
| 1st | #78 | 8 | 6 | 1 | 2 | 17 | 81.0% |
| 2nd | #31 | 4 | 7 | 2 | 2 | 15 | 71.4% |
| 3rd | #59 | 6 | 8 | 1 | 0 | 15 | 71.4% |
| 4th | #56 | 0 | 8 | 2 | 3 | 13 | 61.9% |
| 5th | #86 | 8 | 5 | 0 | 0 | 13 | 61.9% |
| 6th | #54 | 2 | 6 | 2 | 2 | 12 | 57.1% |
| 7th | #77 | 4 | 6 | 1 | 0 | 11 | 52.4% |
| 8th | #92 | 4 | 2 | 2 | 3 | 11 | 52.4% |
| 9th | #82 | 4 | 4 | 0 | 2 | 10 | 47.6% |
| 10th | #79 | 2 | 4 | 1 | 2 | 9 | 42.9% |
| 11th | #32 | 0 | 8 | 0 | 0 | 8 | 38.1% |
| 12th | #83 | 6 | 2 | 0 | 0 | 8 | 38.1% |
| 13th | #135 | 8 | 0 | 0 | 0 | 8 | 38.1% |
| 14th | #138 | 8 | 0 | 0 | 0 | 8 | 38.1% |
| 15th | #139 | 8 | 0 | 0 | 0 | 8 | 38.1% |
| 16th | #142 | 8 | 0 | 0 | 0 | 8 | 38.1% |
| 17th | #143 | 7 | 1 | 0 | 0 | 8 | 38.1% |
| 18th | #146 | 8 | 0 | 0 | 0 | 8 | 38.1% |
| 19th | #147 | 8 | 0 | 0 | 0 | 8 | 38.1% |
| 20th | #81 | 6 | 1 | 0 | 0 | 7 | 33.3% |

## Discussion

In this work, we proposed a fully data-driven ML approach to extract knowledge about which variables are the most informative predictive factors for SARS-CoV-2 pneumonia severity, via FS for data dimensionality reduction. A myriad of works in literature have proposed ML-based algorithms to predict various clinical outcomes in the context of COVID-19: diagnosis, prognosis of clinical evolution, assessments for risks of death, etc. A limited number of them (see Related works) specifically reported on the importance of the features selected, although solely in terms of their predictive power. In this regard, to the best of our knowledge, our work is the first in conducting a systematic and exhaustive study about the inherent properties of the FS procedure itself with COVID-19 data.

We examined a total of 166 FS scenarios, which encompassed all families of FS algorithms: 46 filters (univariate as well as multivariate), 100 wrappers (with deterministic as well as ran-domized search heuristics, and 'wrapped' ML models of different types), and 20 embeddeds. The problem of ordinal classification for severity levels was tackled both as a classification and as a regression task [51]. In particular, via bootstrap sampling techniques, for each FS scenario we evaluated its robustness –or consistency– both in: *a*) an internal manner: stability (i.e. whether the choice of features remained despite changes in the data distribution), and in *b*) an external manner: similarity and common features (to judge the degree of consensus between approaches).

Our motivating dataset consisted of a cohort with $n = 1548$ patients, enrolled in four hospitals from three distinct territories in Spain during the first pandemic wave of COVID-19 (February–May 2020). After quality assurance and pre-processing, the dataset contained $d = 148$

dimensions/features, corresponding to baseline demographic attributes and clinical biofactors recorded at hospital admission, along with sociodemographic information and chronic/acute exposure to different air pollutants at each patient's postcode of residence.

### Strengths: Independent scientific evidence in agreement

Remarkably, we found –*a posteriori*– various independent studies reporting on the significance of the explanatory factors identified in Feature Selection: Selected features. For extensive reviews (beyond the scope of this work), the interested reader could be referred to [74, 75].

Table 6 points out C-reactive protein (CRP) levels as the single most informative magnitude in our study. High CRP concentrations are a common biomarker for infection or systemic inflammation (bacterial, viral or in general), and –already since the early periods of the pandemic– they were described to be in direct association with severe SARS-CoV-2 pneumonia, critical disease development and mortality [74–76], as well as with other major lesions: lungs [77], thrombo-embolism or kidney injury in COVID-19 [78].

Furthermore, PSI was consistently selected by 15 out of our 21 stable FS algorithms. This score, which was first proposed in 1997 to stratify risks in community-acquired pneumonia [79], was found to behave as a good predictor: for either 30-day mortality [80], or 14-day mortality and severity of COVID-19 [81, 82]. For our cohort, age exhibited a notably strong correlation with PSI score (Spearman's $\rho$: mean 0.7854, 95% CI [0.7635, 0.8054], $p<0.001$). This may contribute to explain why age, which has also been often reported as a relevant predictor for different adverse clinical outcomes, was not picked by our FS procedures.

Besides, patients' respiratory status and oxygenation played a prominent role as explanatory information for severity prognosis. As many as four features in this regard were placed in the top-10 ranking from Table 6: $SpO_2$ pulsioximetry, respiratory rate (RR), its ratio $SpO_2$/RR and $SatO_2$/$FiO_2$ from arterial blood gas tests. This is in agreement with [83] –who concluded that $SpO_2<90\%$ was a strong predictor of COVID-19 in-hospital mortality–, or with [4] –who found RR to be a significant predictor for major clinical deterioration: admission in intensive care unit (ICU), or death–. Various works adhering to ML methodologies also found oxygen saturation [2, 6–8, 11] and/or RR [2, 6, 7] among their selected predictors.

The neutrophil-to-lymphocyte ratio (NLR) inflammatory biomarker arose, in our analyses, as one of the most informative and consistent factors to predict SARS-CoV-2 pneumonia severity. Indeed, various studies found NLR to be a strong predictor for adverse outcomes in the context of respiratory diseases (but also for sepsis, cancer, etc. [84]): from 30-day mortality in community-acquired pneumonia [85], to hazard for ICU admission due to COVID-19 [86], or hazard for COVID-19 mortality (overall and by various types of complications) [87]. In this regard, the NLR reflects two dual aspects of the immune system: innate immunity (mostly associated with neutrophils), versus adaptive immunity (mainly lymphocytes) [84]. Complementarily (but with lower ranking), the hematological counts of neutrophils and lymphocytes –addressed separately– were also among the features being selected with moderate repeatability, in alignmnent with the findings in [74].

Interestingly, acute exposures of air pollutants –in particular $NO_2$, $SO_2$, CO and $O_3^-$ were picked as relevant by several of our MI-based FS filters (see Table 6). Independent studies also reported air pollution levels to be meaningful for various clinical outcomes in the context of COVID-19: In [18], it was suggested that $NO_2$ and $PM_{2.5}$ may increase the susceptibility to infection and mortality from COVID-19, whereas [19] reported $NO_2$ exposures to cause severe forms of SARS-CoV-2. Besides, [88] showed high CO concentrations to be positively correlated with COVID-19's reproductive ratio $R_0$, whereas [89] observed positive correlations among the mean values of $PM_{10}$, $NO_2$, CO, and $SO_2$ and the number of COVID-19 cases,

mortality rates and critical cases. A review [90] suggested biological mechanisms for the exposure to air pollutants (such as CO, $NO_2$, $O_3$, $PM_{2.5}$, $PM_{10}$, and $SO_2$) to increase the risk of contracting the SARS-CoV-2 virus. Besides, [91] found that –among different pollutants– $NO_2$, $PM_{2.5}$, $PM_{10}$, and $O_3$ had the strongest correlation with the infection risk of COVID-19. Furthermore, the review by [21] found significant associations between the chronic exposures to $PM_{2.5}$, $PM_{10}$, O3, $NO_2$, $SO_2$ and CO and the incidence, severity and mortality of COVID-19. Conversely, another review [22] mentioned associations between $PM_{2.5}$ and NO with COVID-19 deaths, although called for further research to better test the hypothesis.

## Limitations

Our cohort belongs entirely to the first wave of the COVID-19 pandemic in Spain, from February to May 2020. With such a choice, we aimed at learning patterns from patients who underwent the disease in a situation as homogeneous as possible: regarding the medical knowledge available about COVID-19 and its treatment, and in terms of the situation at the healthcare system. In this regard, we consider interesting for further research to investigate the algorithmic adaptations needed for the FS models to accommodate datasets with time-induced distributional shifts [92, 93] (i.e. data or trends changing across pandemic waves).

This first-wave situation was also detrimental for the process of collecting clinical data, and for their integrity. The clinical members in our research team were responsible for attending an unprecedented therapeutic demand, with shortage of personnel and resources. These extraordinarily difficult circumstances, in which the clinical information for our study was collected, may explain –arguably, to a major extent– the high rates of missing values in our dataset.

Due to this limitation in the availability of valid measurements, we were forced to discard 14 variables with ≥60% missingness. Namely: SOFA score, along with AST, bilirubin, CPK, IL-6, BNP, troponin, ferritin, eosinophils, and platelets from blood tests, and with pH, $PaO_2$, $PaCO_2$, and $PaO_2/FiO_2$ from arterial blood gas tests. Thus, as a direct consequence, here we were precluded from performing any analyses or extracting any conclusions about these 14 factors.

In particular, significant associations were reported between lower Pa O2/Fi O2 values and adverse clinical outcomes due to COVID-19 (admission to ICU) [86, 94]. A comparable situation arises for platelets: the Systematic Inflammation Index (SII) [95] –defined as the product between NLR and platelet count– was recently reported as a robust predictor for the need of intubation, but our cohort lacked of sufficient valid data (a remarkably low fraction, even) as to consider the role of platelets. In the same manner, the Model for Early COvid-19 Recognition (MECOR) [96] has become an emerging score for prompt COVID-19 diagnostic triage in patients with community-acquired pneumonia. However, given that it combines blood counts of leukocytes, lymphocytes, monocytes, neutophils and platelets, it was also impossible for us to incorporate MECOR in our analyses. Similar reasonings hold for ferritin [97], IL-6 [74, 98], AST [74] or troponin [74], etc.

From a data perspective, this issue of prevailing data missingness forced us to incorporate imputation strategies (*k*nn, iterative) which might have altered the informativeness of features. Interestingly, RBA filters –which are able to handle missing values natively– tended, in general, to show better properties in terms of stability.

A certain degree of inclusion bias may have also been incorporated in our cohort, particularly by the admission policies at hospital *C*: forced by the unprecedented situation of pandemic, and considering that such institution had more ICU beds than other hospitals in its region, patients who –during preliminary examinations– were triaged as more fragile or

deteriorated, were preferentially referred there. This issue could explain, at least to an important extent, the larger portion of severe cases admitted at hospital *C*.

Moreover, a subset of the features are not patient-specific, but instead related to his/her postcode of residence: the 7 socioeconomic attributes, as well as the 32 features describing chronic and acute exposures to air pollutants. Patients from the same hospital (or geographical area) tended to have similar values, hence arguably introducing a latent correlation with respect to the distribution of severities at each location.

## Conclusions

This work covers a systematic and exhaustive exploration of FS techniques aimed at discovering the most informative predictor variables for SARS-CoV-2 pneumonia severity in our clinical dataset. We made particular emphasis on the relevance of our results and the features selected, by means of objective criteria regarding the stability of each method (bootstrapped), as well as the similarity across them (Jaccard index of agreement, frequencies per feature).

Out of the different 166 FS scenarios evaluated, 21 achieved robust stability. Attending to the similarities of selected subsets, our fully data-driven strategy identified a number of variables as informative. These consistently included: CRP, PSI, respiratory rate (RR) and oxygen levels ($SpO_2$, $SpO_2/RR$, $SatO_2/FiO_2$), NLR, LDH, and PCT.

Remarkable agreement has been found *a posteriori* with independent clinical research on COVID-19 patient outcomes, hence stressing the suitability of this type of data-driven approaches for knowledge extraction, to support pulmonologists regarding risk factors for COVID-19 severity.

## Supporting information

**S1 Appendix. On-line supplementary materials.** Report—(S.A) Data: List of features, (S.B) Data: Cohort characteristics, (S.C) Methods: Hyperparameters, (S.D) Results: Stability, (S.E) Results: Computation times.
(PDF)

## Acknowledgments

We would like to acknowledge patients who participated in this research, as well as the staff at the four hospitals involved: *Hospital Clínic i Provincial de Barcelona*, *Hospital Universitari i Politècnic La Fe de Valencia*, *Galdakao-Usansoloko Unibertsitate Ospitalea* and *Gurutzetako Unibertsitate Ospitalea*. We are particularly grateful to all members of the COVID-19 & Air Pollution Working Group.

### Members of the COVID-19 & Air Pollution Working Group

**La Fe University and Polytechnic Hospital, Pneumology Department**: Ana Latorre, Paula González Jiménez, Raul Méndez, Rosario Menéndez.

**Cruces University Hospital, Pneumology Service**: Leyre Serrano Fernández, Eva Tabernero Huguet, Luis Alberto Ruiz Iturriaga, Rafael Zalacain Jorge.

**Hospital Clínic of Barcelona, Pneumology Department**: Antoni Torres, Catia Cilloniz.

**Galdakao-Usansolo University Hospital, Respiratory Service**: Pedro Pablo España Yandiola, Ana Uranga Echeverría, Olaia Bronte Moreno, Isabel Urrutia Landa.

**Galdakao-Usansolo University Hospital, Research Unit**: Jose María Quintana, Susana García-Gutiérrez, Mónica Nieves Ermecheo, María Gascón Pérez, Ane Villanueva.

**BioCruces Bizkaia Health Research Institute**: Mónica Nieves Ermecheo.

**Basque Center for Applied Mathematics (BCAM)**: Dae-Jin Lee, Fernando García-García, Miren Hayet-Otero, Inmaculada Arostegui.

**University of the Basque Country (UPV/EHU), Department of Mathematics**: Inmaculada Arostegui.

**Universitat Politècnica de València, Department of Applied Statistics and Operational Research, and Quality**: Joaquín Martínez-Minaya.

## Code sharing

The Python code developed for our analyses is publicly available on our GitHub repository: https://github.com/fegarcia-bcam/FeatSel-COVID-19-PLOS-ONE.

## Copyright statement

S9 Fig in S1 Appendix (geographical distribution of hospitalized cases in our cohort, in total and by SARS-CoV-2 pneumonia severity), S10 and S11 Figs in S1 Appendix (socioeconomic data), and S12–S19 Figs in S1 Appendix (exposure to air pollution) all depict maps for data elaborated within the context of this research (see section Clinical data collection for further details). They were plotted with `geopandas` software [69] and public postcode polygons for Spain (CartoCiudad, CC BY 4.0 www.scne.es/productos.html#CartoCiudad).

## Author Contributions

**Conceptualization:** Miren Hayet-Otero, Fernando García-García, Dae-Jin Lee, Inmaculada Arostegui.

**Data curation:** Fernando García-García, Pedro Pablo España Yandiola, Isabel Urrutia Landa, Mónica Nieves Ermecheo, Rosario Menéndez, Antoni Torres, Rafael Zalacain Jorge.

**Formal analysis:** Miren Hayet-Otero, Fernando García-García, Joaquín Martínez-Minaya.

**Funding acquisition:** Dae-Jin Lee, Pedro Pablo España Yandiola.

**Investigation:** Miren Hayet-Otero, Fernando García-García, Dae-Jin Lee, Joaquín Martínez-Minaya, Pedro Pablo España Yandiola, Isabel Urrutia Landa, Mónica Nieves Ermecheo, José María Quintana.

**Methodology:** Miren Hayet-Otero, Fernando García-García, Dae-Jin Lee, Joaquín Martínez-Minaya.

**Project administration:** Dae-Jin Lee, Pedro Pablo España Yandiola, Isabel Urrutia Landa.

**Resources:** Pedro Pablo España Yandiola, Isabel Urrutia Landa.

**Software:** Miren Hayet-Otero, Fernando García-García.

**Supervision:** Dae-Jin Lee, José María Quintana, Inmaculada Arostegui.

**Validation:** Fernando García-García.

**Visualization:** Miren Hayet-Otero, Fernando García-García, Joaquín Martínez-Minaya.

**Writing – original draft:** Miren Hayet-Otero, Fernando García-García.

**Writing – review & editing:** Dae-Jin Lee.

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
