## [Decision Letter · Decision Letter 0]

6 Nov 2022

PONE-D-22-28067Extracting relevant predictive variables for COVID-19 severity prognosis: An exhaustive comparison of feature selection techniquesPLOS ONE

Dear Dr. García-García,

Thank you for submitting your manuscript to PLOS ONE. After careful consideration, we feel that it has merit but does not fully meet PLOS ONE’s publication criteria as it currently stands. Therefore, we invite you to submit a revised version of the manuscript that addresses the points raised during the review process.

We look forward to receiving your revised manuscript.

Kind regards,

Sathishkumar V E

Academic Editor

PLOS ONE

“We would like to acknowledge patients who participated in this research, as well as the 545

staff at the four hospitals involved: Hospital Cl´ınic i Provincial de Barcelona, Hospital

Universitari i Polit`ecnic La Fe de Valencia, Galdakao-Usansoloko Unibertsitate

Ospitalea and Gurutzetako Unibertsitate Ospitalea. We are particularly grateful to all

members of the COVID-19 & Air Pollution Working Group.

This research is partially supported by the Basque Government through the grant 550

Artificial Intelligence in BCAM 2019/00432, the ‘Mathematical Modelling Applied to

Health’ strategy, and the BERC 2018–2021 & 2022–2025 programmes; by the Spanish

Ministry of Science, Innovation and Universities under BCAM Severo Ochoa

accreditation SEV-2017-0718, as well as by the Spanish State Research Agency (AEI)

through project S3M1P4R (PID2020-115882RB-I00 ).”

“This research is partially supported by the Basque Government through the grant Artificial Intelligence in BCAM 2019/00432, the ‘Mathematical Modelling Applied to Health’ strategy, and the BERC 2018–2021 & 2022–2025 programmes; by the Spanish Ministry of Science, Innovation and Universities under BCAM Severo Ochoa accreditation SEV-2017-0718, as well as by the Spanish State Research Agency (AEI) through project S3M1P4R (PID2020-115882RB-I00).”

6. We note that Supporting Figures S9-S19 in your submission contain [map/satellite] images which may be copyrighted. All PLOS content is published under the Creative Commons Attribution License (CC BY 4.0), which means that the manuscript, images, and Supporting Information files will be freely available online, and any third party is permitted to access, download, copy, distribute, and use these materials in any way, even commercially, with proper attribution. For these reasons, we cannot publish previously copyrighted maps or satellite images created using proprietary data, such as Google software (Google Maps, Street View, and Earth). For more information, see our copyright guidelines: http://journals.plos.org/plosone/s/licenses-and-copyright.

a. You may seek permission from the original copyright holder of Figures S9-S19 to publish the content specifically under the CC BY 4.0 license. 

Reviewers' comments:

Reviewer's Responses to Questions

**Comments to the Author**

1. Is the manuscript technically sound, and do the data support the conclusions?

Reviewer #1: No

Reviewer #2: Yes

2. Has the statistical analysis been performed appropriately and rigorously? 

Reviewer #1: Yes

Reviewer #2: Yes

3. Have the authors made all data underlying the findings in their manuscript fully available?

Reviewer #1: No

Reviewer #2: Yes

4. Is the manuscript presented in an intelligible fashion and written in standard English?

Reviewer #1: Yes

Reviewer #2: Yes

5. Review Comments to the Author

Reviewer #1: This paper by Hayet-Otero et al. proposes a complex mathematical model for a fully data driven approach, assessing the relevant predictors of severity in Covid-19 disease. Although this topic looks interesting, the method used to fill biofactors and clinical data into the mathematical model could result questionable, in that it did not consider some previous reports on prognostic predictors of both community acquired pneumonia (CAP) and Covid-19 disease.

Specific Comments

1. Neutrophil-to-Lymphocyte Ratio was shown to predict intra-hospital mortality in patients with CAP (Cataudella et al., J Am Ger Soc 2017), performing better than PSI and CURB-65. This evidence further strenghtened that combined, but not separate, evaluation of neutrophils and lymphocytes should be considered mandatory to improve prognosis prediction. This is due to the fact that in patients with worse prognosis neutrophilia runs in parallel with lymphopenia.It does not seem that Authors follow this strategy, since they used absolute values of both cells separately.

2. A poorer prognosis of Covid-19 patients strictly depends on an early fall in PaO2/FiO2, that was recently shown to be inversely related to NLR (Regolo et al., J Clin Med 2022). It looks as if Authors filled in the model only O2 levels. What about PaO2/FiO2?

3. Systemic Inflammation Index (SII), a combined index using neutropohils, platelets and lymphocytes, was shown to be another important predictor of intubation in Covid-19 patients (Muhammed et al., Pathogens 2021). Did Authors consideri t?

4. MECOR is an emerging score to recognize a possible viral etiology of CAP, providing a prompt Covid-19 triage (Sambataro et al., Diagnostics 2020), helping to prevent complications.

5. So far, Covid 19 disease is considered the result of a derangement between innate and adaptive immunity (for review, see Buonacera et al., Int J Mol Sci 2022). This is a furher reason for paying attention to the strategy used in filling biofactors and clinical data into the mathematical model (see points 1,2,3)

Reviewer #2: 1.Introduction section needs to be re-written to improve its quality and readability.

2.The literature has to be strongly updated with some relevant and recent papers focused on the fields dealt with in the manuscript.

3. Clinical data can be summarized in a table for better understanding.

4. Explain in detail what type of hyperpaprameter selection methodology is used for selecting best parameters for each ML algorithms.

5. Explain why the current method was selected for the study, its importance and compare with traditional methods.

6.Authors are suggested to include more discussion on the results and also include some explanation regarding the justification to support why the proposed method is better in comparison towards other methods

7.Does this kind of study have never attempted before? Justify this statement and give an appropriate explanation to do so in this paper.

8. There are several machine learning algorithms what is the reason for using onlt LR, Ridge and KNN?.

9. Mentioning number of features before and after using FS algorithms in table can be a simplified version of summarizing.

10. Overall the presentation and provided results are satifactory.

6. PLOS authors have the option to publish the peer review history of their article (what does this mean?). If published, this will include your full peer review and any attached files.

Reviewer #1: No

Reviewer #2: **Yes: **Usha Moorthy

---

## [Author Response · Author response to Decision Letter 0]

16 Mar 2023

Please find attached an extra document entitled 'Response to Reviewers.pdf', which contains a point-by-point response to the questions and remarks stated by each of them.

---

## [Decision Letter · Decision Letter 1]

27 Mar 2023

Extracting relevant predictive variables for COVID-19 severity prognosis: An exhaustive comparison of feature selection techniques

PONE-D-22-28067R1

Dear Dr. García-García,

We’re pleased to inform you that your manuscript has been judged scientifically suitable for publication and will be formally accepted for publication once it meets all outstanding technical requirements.

Kind regards,

Sathishkumar V E

Academic Editor

PLOS ONE

Additional Editor Comments (optional):

Reviewers' comments:

Reviewer's Responses to Questions

**Comments to the Author**

1. If the authors have adequately addressed your comments raised in a previous round of review and you feel that this manuscript is now acceptable for publication, you may indicate that here to bypass the “Comments to the Author” section, enter your conflict of interest statement in the “Confidential to Editor” section, and submit your "Accept" recommendation.

Reviewer #1: All comments have been addressed

Reviewer #2: (No Response)

2. Is the manuscript technically sound, and do the data support the conclusions?

Reviewer #1: Yes

Reviewer #2: (No Response)

3. Has the statistical analysis been performed appropriately and rigorously? 

Reviewer #1: Yes

Reviewer #2: (No Response)

4. Have the authors made all data underlying the findings in their manuscript fully available?

Reviewer #1: Yes

Reviewer #2: (No Response)

5. Is the manuscript presented in an intelligible fashion and written in standard English?

Reviewer #1: Yes

Reviewer #2: (No Response)

6. Review Comments to the Author

Reviewer #1: This Reviewer has no further concern, since all comments were fulfilled. This paper has been improved.

Reviewer #2: (No Response)

7. PLOS authors have the option to publish the peer review history of their article (what does this mean?). If published, this will include your full peer review and any attached files.

Reviewer #1: No

Reviewer #2: **Yes: **Usha Moorthy

---

## [Editor Report · Acceptance letter]

30 Mar 2023

PONE-D-22-28067R1 

Extracting relevant predictive variables for COVID-19 severity prognosis: An exhaustive comparison of feature selection techniques 

Dear Dr. García-García:

I'm pleased to inform you that your manuscript has been deemed suitable for publication in PLOS ONE. Congratulations! Your manuscript is now with our production department. 

Kind regards, 

on behalf of

Dr. Sathishkumar V E 

Academic Editor

PLOS ONE